

# Effect of aerosol sub-grid variability on aerosol optical depth and cloud condensation nuclei: Implications for global aerosol modelling.

N. Weigum[1], N. Schutgens[1], and P. Stier[1]

[1]Department of Physics, University of Oxford, Oxford, UK

*Correspondence to:* N. Weigum
(weigum@atm.ox.ac.uk)

**Abstract.** A fundamental limitation of grid-based models is their inability to resolve variability on scales smaller than a grid box. Past research has shown that significant aerosol variability exists on scales smaller than these grid-boxes,which can lead to discrepancies in simulated aerosol climate effects between high and low resolution models. This study investigates the impact of neglecting sub-

grid variability in present-day global microphysical aerosol models on aerosol optical depth (AOD) and cloud condensation nuclei (CCN). We introduce a novel technique to isolate the effect of aerosol variability from other sources of model variability by varying the resolution of aerosol and trace gas fields while maintaining a constant resolution in the rest of the model.

We compare WRF-Chem runs in which aerosol and gases are simulated at 80 km and again at 10

km resolutions; in both simulations the other model components, such as meteorology and dynamics, are kept at the 10 km baseline resolution. We find that AOD is underestimated by 13% and CCN is overestimated by 27% when aerosol and gases are simulated at 80 km resolution compared to 10 km. Processes most affected by neglecting aerosol sub-grid variability are gas-phase chemistry and aerosol uptake of water through aerosol/gas equilibrium reactions. The inherent non-linearities

in these processes result in large changes in aerosol properties when aerosol and gaseous species are artificially mixed over large spatial scales. These changes in aerosol and gas concentrations are exaggerated by convective transport, which transports these altered concentrations to altitudes where their effect is more pronounced. These results demonstrate that aerosol variability can have a large impact on simulating aerosol climate effects, even when meteorology and dynamics are held

constant. Future aerosol model development should focus on accounting for the effect of sub-grid variability on these processes at global scales in order to improve model predictions of the aerosol effect on climate.





## 1  Introduction

Aerosols are known to have a significant effect on the earth's climate through their interactions with
radiation and clouds. Aerosols interact with incoming solar radiation by scattering and absorption,
resulting a net cooling of the Earth (Boucher et al., 2013). Absorption can also cause a number of
rapid adjustments to the climate system through local heating of the atmosphere (Koch and Del Ge-
nio, 2010). Aerosols interact with clouds by serving as cloud condensation nuclei (CCN) and/or ice
nuclei (IN). The number of CCN can affect cloud radiative properties thereby altering cloud albedo
(Twomey, 1974). Additionally, aerosols acting as CCN are hypothesized to affect precipitation ef-
ficiency, cloud lifetime, and cloud thickness, although these interactions are complex and uncertain
(Albrecht, 1989; Rosenfeld et al., 2008). The total effective radiative forcing due to aerosols in-
cluding both radiation and cloud interactions is estimated to be -0.9 (-1.9 to -0.1) W m$^{-2}$ (Boucher
et al., 2013), which counteracts approximately one-third of the positive radiative forcing caused by
greenhouse gases. Aerosols continue to contribute the largest uncertainty to estimates of the Earth's
energy budget (Boucher et al., 2013).

Prediction of the aerosol effect on climate depends on the ability of global climate models (GCM)
to accurately estimate aerosol concentrations and their microphysical properties. However, a funda-
mental limitation of grid-based GCMs is their inability to capture spatial variations smaller than the
size of their grid boxes, which typically range from 100 – 400 km for aerosol climate simulations.
Significant aerosol variability exists on scales smaller than global climate model grid-boxes (e.g.
Anderson et al., 2003; Weigum et al., 2012), and discrepancies between aerosol modelling schemes
and observations have been attributed to these sub-grid spatial variations (e.g. Gustafson et al., 2011;
Benkovitz and Schwartz, 1997). It is therefore important to determine the extent to which different
sub-grid scale processes contribute to the discrepancies in aerosol modelling in order to focus model
development on improving parameterisations of these important aerosol processes.

Previous studies have explored the effect of neglecting sub-grid aerosol variability on simulations
of aerosol fields. Most of these studies address this issue by varying model resolution and evaluat-
ing the subsequent effect on aerosol fields. Gustafson et al. (2011) compared regional-scale model
simulations at 75 km and 3 km resolutions to quantify the error introduced from neglected sub-grid
variability on shortwave direct aerosol radiative forcing. They found an average mean bias of over
30% in the 75 km simulation compared to the 3km simulation.Wainwright et al. (2012) investigated
the effect of model resolution on secondary organic aerosol concentrations and found that summer-
time predictions increased by 20-30% at higher resolutions. Metzger et al. (2002) studied the impact
of changing a global climate model's resolution on equilibrium concentrations of aerosol nitrate,
finding an overestimation of 30-80% at low resolutions. There exist numerous further examples of
studies that vary model resolution and evaluate the subsequent impact on aerosol-cloud interactions,
aerosol radiative forcing, and precipitation (e.g. Myhre et al., 2002; Ekman and Rodhe, 2003; Owen
and Steiner, 2012).



In these studies, simulated aerosols are affected by changes in resolution of a multitude of different meteorological, dynamical, and microphysical fields, making it difficult to isolate and understand the impact of a particular aerosol process. Because the focus in our study is on the importance of aerosol variability, it is crucial to disentangle aerosol variability from other sources of variability within the model.

We developed a technique to simulate aerosol processes at varying resolutions while maintaining a constant resolution in all other model fields. While the proposed technique has not yet been applied in the context of aerosol variability, there have been previous attempts to run different model components at varying resolutions. These studies have mostly focused on separating the dynamical core of the model from the physical parameterisations to test the resolution convergence of the dynamical
core (Held and Suarez, 1994; Williamson, 1999).

A more recent study examined the resolution dependence of cloud microphysics parameterisations by holding the resolution of the dynamics grid constant and changing the grid spacing of the selected parameterisations (Gustafson et al., 2013). In this set-up, their model ran at a specified fine-scale resolution, which communicated at each time step with another copy of the model physics
on an alternate, coarse resolution grid. This was done by coarsening the fine-scale dynamics to the alternative grid for additional coarse-grid physics calculations that were not permitted to feed back into the fine dynamics grid. However, even though the original dynamics grid was not altered by the coarse-grid physics, the coarse-grid physics could only interact with the coarsened version of the dynamics and vice versa (fine-scale physics with fine-scale dynamics).

The above methods required modifications to the original model and, as a result, potentially introduced differences between coarse-grid and fine-grid components arising from factors other than resolution changes. In this paper, we present a method that offers an alternative approach to varying the resolution of different model components separately from one another, specifically varying the resolution of aerosol and gas processes separately from the physics and dynamics of the model. We
use two grids, one coarse and one fine; however, in our setup, both the fine-grid and coarse-grid aerosols and gases interact with the fine-grid meteorology and dynamics, so that any differences in the simulations are due solely to changes in aerosol variability. In Section 2, we describe the implementation of this technique, as well as the model configuration and grid set-up. The results are presented in three sections. Sections 3.1 and 3.2 explore the impact of neglecting aerosol sub-grid
variability on AOD and CCN by varying the resolution of aerosols and trace gases separately from the rest of the model. Section 3.3 compares the results of the previous two sections to simulations where the resolution of the entire model is varied, as done in traditional model resolution comparisons in order to demonstrate the difficulty in separating meteorological and aerosol effects in traditional model resolution studies. Finally, the results are discussed and summarised in Section 4.





## 2 Methods

### 2.1 Experimental Design

In order to understand how sub-grid aerosol variability affects model predictions of aerosol fields, we modify the chemistry version of the Weather and Research Forecast model (WRF-Chem) (Grell et al., 2005) so that it is capable of simulating aerosol microphysical processes at a different resolution than the dynamical and meteorological processes. The purpose of this technique is to recreate the artificial mixing of trace gas and aerosol properties that occurs in global climate models, while maintaining a constant resolution in the other fields within the model. This is accomplished by running the model at a specified high resolution and averaging the aerosol and trace gas fields online over a pre-defined, lower resolution grid.

Figure 1 describes the process conceptually. The grid in Figure 1a represents the high resolution aerosol and gas fields. To simulate these fields at a lower resolution than the rest of the model, we take the mean value of all of the high resolution grid cells residing within the corresponding low resolution grid cell and re-assign each of the high resolution cells to the mean value, as depicted in Figure 1b. This occurs after each aerosol process. This means that even though the aerosol and trace gas species are calculated on the high resolution grid, each fine grid cell within the coarse grid cell has the same value. Therefore, from the model's perspective, the fields are equivalent to a low resolution grid similar to Figure 1c.

The modular structure of WRF-Chem allows for easy execution of this experimental design. In WRF-Chem, the aerosol and gas-phase processes occur within the "chemistry driver", which contains separate modules for each aerosol process. These modules include emissions, photolysis, dry deposition, vertical mixing and wet deposition by convective transport, gas-phase chemistry, and aerosol microphysical processes. In our modified set-up, the aerosol and gaseous fields are averaged over the lower resolution grid before and after each module within the chemistry driver so that every time the aerosol and gaseous fields are modified, their concentrations are once again averaged over the low resolution grid. The averaged fields are then passed onto the rest of the model. This process is repeated at every time step. As a result, the aerosol and gaseous species are effectively simulated at a lower resolution while allowing for interaction with the high resolution meteorology.

With this design, the resolution of the aerosol and gaseous fields can be varied by simply changing the number of high resolution grid points over which the fields are averaged. In this paper, we refer to these types of simulations as "aerosol averaged" (AA) runs. These aerosol averaged runs can then be compared to simulations in which the aerosols are simulated at the same resolution as the rest of the model. We refer to these simulations as "full resolution aerosol" (FRA) runs.



### 2.1.1   Grid Set-up

The study is conducted over a 1,280 km by 1,280 km grid, encompassing nearly all of the United
Kingdom and north-western France. To prevent unrealistic interactions between the averaged fields
and the boundary conditions, we apply the "aerosol averaging" technique only to the inner 640 km
x 640 km grid. We limit the analysis to this region, which covers the southern half of the United
Kingdom and the English Channel. The left panel of Figure 2 shows the terrain height of the entire
outer grid, with the inner analysis region outlined in the centre of the grid. The right panel shows
the average hourly ammonia emissions, providing an example of how the averaging technique is
applied to the inner grid only (emissions are kept at high resolution; this diagram is for visualisation
purposes only).

We conduct the baseline high resolution simulation at 10 km. In this run, all fields (i.e. aerosols,
dynamics, meteorology) are simulated at 10 km resolution, and it is referred to as FRA10. We chose
10 km as it is the highest recommended resolution WRF-Chem can run with a convective parame-
terisation (Gerard, 2007).

To determine how the sub-grid variability of aerosol processes impacts model predictions of im-
portant aerosol properties, we conduct three "aerosol averaged" runs during which the aerosol res-
olution is set to 40 km, 80 km, and 160 km, while maintaining a resolution of 10 km in all other
model fields. We refer to these runs as AA40, AA80 and AA160, respectively.

The majority of our analysis focuses on comparisons between FRA10 and AA80. The grid spacing
of AA80 is representative of the maximum resolution at which aerosols can be simulated in current
GCMs for climate simulation purposes, and therefore demonstrates the degree of aerosol variability
that these models are able to capture. We use the results of AA40 and AA160 to show the effect
of increasing and decreasing the resolution of aerosol processes with respect to this current GCM
resolution. In addition to these simulations, we conduct a second full resolution simulation at a res-
olution of 80 km (FRA80). We also compare this simulation to FRA10 in order to demonstrate how
traditional resolution comparison studies miss important information due to their inability to sepa-
rate aerosol and meteorological effects. Table 1 summarises the different WRF-Chem simulations
analysed in this paper.

The model simulations are conducted for one month from May 1 - 31, 2008. The first two days
are used as a spin-up period; therefore, the analysis is carried out over the period from May 3 - 31,
2008.

### 2.2   Model Configuration

This study uses version 3.3.1 of WRF-Chem Grell et al. (2005) and Fast et al. (2006). The meteo-
rological model WRF (Skamarock and Klemp, 2008) is a mesoscale numerical weather prediction
system designed for both operational forecasting and atmospheric research purposes across scales



ranging from metres to hundreds of kilometres. WRF-Chem provides a number of options for gas-phase chemistry and aerosol processes, including biogenic and anthropogenic emissions, dry and wet

deposition, photolysis, vertical turbulent mixing, gas and aqueous phase chemical transformation, aerosol chemistry and microphysics as well as aerosol direct and indirect effects through interaction with atmospheric radiation and cloud microphysics. The main options for the physical and chemical schemes employed in the simulations are summarised in Table 2.

The aerosol module used in this analysis is the MADE/SORGAM module, consisting of the Modal

Aerosol Dynamics Model for Europe (MADE), which handles the inorganic and primary organic constituents, and the Secondary Organic Aerosol Module (SORGAM), which handles the secondary organic fraction (Ackermann et al., 1998; Schell et al., 2001). In MADE/SORGAM, the aerosol size distribution is described by three overlapping modes, representing the Aitken, accumulation, and coarse modes. The distribution within each mode is assumed to be log-normal with fixed standard

deviations of 1.7, 2.0 and 2.5 for the Aitken, accumulation, and coarse modes, respectively (Ackermann et al., 1998).

The aerosol species treated in MADE/SORGAM are ammonium ($NH_4^+$), nitrate ($NO_3^-$), sulphate ($SO_4^{2-}$), elemental carbon (EC), organic matter (OM, primary and secondary), aerosol water, sea salt, and mineral dust. The processes treated are homogeneous nucleation in the sulphuric acid-

water system, condensation of sulphuric acid vapour, and coagulation by Brownian motion. Aerosol water uptake and formation of nitrate and ammonium is determined through the ammonia/nitric acid/sulphuric acid thermodynamic equilibrium system, which is parameterised based on the Model for an Aerosol Reacting System (MARS) (Saxena et al., 1986). Photolysis rates are simulated by the Fast-J scheme (Wild et al., 2000), and the dry deposition velocities are determined by the We-

sely parameterisation (Wesely, 1989). Wet deposition is handled in a simplified parameterisation of convective updrafts for trace gases and inorganic aerosols. There is currently available a full wet deposition module coupled with aqueous chemistry; however, in WRF-Chem these options are only available when aerosol radiative feedback is turned on. Because the aim of this experiment is to compare simulations with identical meteorology, aerosol feedback to the radiation schemes must be

switched off. Without wet deposition due to large-scale precipitation, a significant removal process is missing, which will likely result in higher aerosol concentrations than if the process were included.

All simulations use identical initial and boundary conditions generated by WRF-Chem from idealised profiles. The values are based on idealised, northern hemispheric, mid-latitude, clean environmental, vertical profiles from the NOAA Aeronomy Lab Regional Oxidant Model (McKeen et al.,

1991). Meteorological boundary conditions were nudged to National Centers for Environmental Protection Final (NCEP FNL) operational global analysis data, which are available every 6 hours on a 1°by 1° grid.





### 2.2.1 Emissions

Anthropogenic emissions are taken from the Netherlands Organization for Applied Scientific Re-
search (TNO), a detailed European gridded emission inventory developed by van der Gon et al.
(2010) in the framework of the European MACC project (http://gmes-atmosphere.eu). The inven-
tory contains high resolution (1/8° lon x 1/6° lat) emissions for $NO_x$, $SO_2$, non-methane volatile
organic compounds (NMVOC), $CH_4$, $NH_3$, CO, $PM_{10}$ and $PM_{2.5}$, which are interpolated to the
WRF domain to give hourly emissions per square kilometre.

The $PM_{2.5}$ emissions are broken into components of organic carbon, elemental carbon, sulphate,
and "other mineral components" using composition profiles developed for the TNO inventory. These
componenst are split into 20% Aitken mode and 80% accumulation. $PM_{10}$ emissions remain unspe-
ciated as coarse mode particulate matter. Total NMVOC emissions are divided into their constituent
RADM2 species to be handled by WRF-Chem.

Biogenic emissions are calculated online with a module based on the parameterisation by Guen-
ther et al. (1994) using the U.S. Geological Survey 24 land use categories provided by the standard
WRF configuration. Sea salt and dust emissions (Shaw et al., 2008) are also calculated online and
are proportional to 10-metre wind speed over salt water for sea-salt and over non-urban land surfaces
with sparse vegetation for dust.

## 3 Results

We present the results in three sections. The first two sections explore the impact of aerosol sub-
grid variability on AOD at 600 nm and CCN at 0.5% supersaturation using the "aerosol averaged"
technique. The third section presents results from the full resolution run at 80 km (FRA80) to demon-
strate the difficulty in separating meteorological and aerosol effects in traditional model resolution
studies.

In all comparisons, the FRA10 simulation is taken as the "truth". The FRA10 simulation is in-
tended to be representative of typical aerosol conditions in the specific environment of the simulation
and is meant to capture most of the aerosol variability important for accurately depicting aerosols'
microphysical evolution and effect on climate.

### 3.1 Effect of aerosol sub-grid variability on AOD

Figure 3 presents results of simulated AOD for the FRA10, AA40, AA80 and AA160 where we vary
the resolution of aerosol and gaseous species from 10 km to 40 km, 80 km, and 160 km, respectively.
We calculated the differences by first coarse-graining the results from the high resolution simulation
to the grid of the low resolution run to which it is being compared. This eliminates differences
due to the inevitably smoother low resolution run not being able to capture the same degree of
variability as the high resolution simulation. We find that at lower aerosol resolutions, simulated



AOD is underestimated with respect to the high resolution run. Relative to FRA10, the negative bias in monthly averaged AOD increases from an average of -9.4%, to -13.1% to -15.8% as the aerosol resolution is decreased to 40 km, 80 km, and 160 km, respectively. We investigate the mechanisms behind this underestimation by exploring differences between the FRA10 and AA80 simulations.

We performed pattern correlation analysis between the hourly spatial differences in AOD in the FRA10 and AA80 simulations and the hourly spatial differences in a number of aerosol properties known to have an impact on AOD. The analysis revealed that differences in AOD between the FRA10 and AA80 simulations are highly correlated to differences in accumulation mode aerosol water content, with an average correlation of 0.97 over the entire time period. Accumulation mode nitrate and ammonium also demonstrate high correlations, with averages of 0.84 and 0.82, respectively.

It is clear that uptake of water by accumulation mode aerosols plays an important role in the underestimation of AOD in the low aerosol resolution runs, as shown in Figure 4. Compared to Figure 3, we can see the strong relationship between the two properties, as confirmed by the correlation analysis.

This is not surprising as many studies have shown that aerosol water content has a large impact on aerosol optical properties. Shinozuka et al. (2007) used aircraft measurements to show that the fraction of ambient AOD due to water uptake is $37 \pm 15\%$ over continental U.S.; the fraction is likely even higher over marine environments. Using a box model, Pilinis et al. (1995), found that in their simulations the most important process in determining aerosol direct radiative forcing is increase in aerosol mass as a result of water uptake. In both the FRA10 and AA80 simulations aerosol water content makes up approximately two thirds of the total aerosol mass, making AOD highly sensitive to changes in water.

### 3.1.1 Investigation of aerosol water uptake in WRF-Chem

In WRF-Chem, the total aerosol water content is calculated using a program based on the Model for an Aerosol Reacting System (MARS) described in Saxena et al. (1986), which determines the amount of water taken up by the complex of sulphate ($SO_4^{2-}$), nitrate ($NO_3^-$), and ammonium ($NH_4^-$) aerosol species. At thermodynamic equilibrium, the amount of water contained in these particles depends on temperature, relative humidity (RH), and aerosol amount and composition, the latter of which, in turn, depends on the concentrations of the gaseous precursors ammonia ($NH_3$), nitric acid vapour ($HNO_3$), and sulphuric acid vapour ($H_2SO_4$) (Seinfeld and Pandis, 2006). We explicitly designed this study so that temperature and relative humidity are identical in both the FRA10 and AA80 simulations; therefore, the changes in aerosol water content must be due to changes in aerosol amount and/or composition.

Although the RH fields are the same in the two runs, different aerosol types react differently at particular levels of RH. Aerosols such as nitrate and ammonia exhibit deliquescent behaviour, with a deliquescent relative humidity (DRH) of approximately 60% (Saxena et al., 1986). Sulphuric acid,





on the other hand, is hygroscopic, meaning it readily absorbs water at nearly all RH and does not display this step-function behaviour in water absorption.

In the sulphate-ammonium-nitrate-water system, the relative amounts of these aerosols are determined by competition between the following two thermodynamic equilibrium reactions (Seinfeld and Pandis, 2006):

$$2\,NH_3(g) + H_2SO_4(g) \rightleftharpoons (NH_4)_2SO_4(aq)$$


$$NH_3(g) + HNO_3(g) \rightleftharpoons NH_4NO_3(aq)$$

In this system, the first reaction dominates; ammonia preferentially neutralises sulphuric acid due to its low saturation vapour pressure and drives the reaction to the aerosol phase. Therefore, ammo-

nium nitrate ($NH_4NO_3$) is formed only when there is sufficient ammonia to neutralise the amount of sulphate present, i.e. in areas of high concentrations of ammonia and/or low concentrations of sulphate. $(NH_4)_2SO_4$ is the preferred form of sulphate, meaning that each mole of sulphate will remove two moles of ammonia from the gas phase. The system is therefore divided into two cases of interest: high-ammonia and low-ammonia.

In the low-ammonia case, there is insufficient $NH_3$ to neutralise the available sulphate. The sulphate present will tend to drive the nitrate to the gas phase. The partial pressure of ammonia is low, resulting in zero or near-zero levels of ammonium nitrate.

In the high-ammonia case, there is excess ammonia so that the aerosol phase is largely neutralised. The ammonia that does not react with sulphate will be available to react with nitric acid vapour to

produce $NH_4NO_3$.

Essentially, at very low ammonia concentrations, ammonium sulphate primarily constitutes the aerosol composition. As ammonia increases, ammonium nitrate becomes a significant aerosol constituent once sulphate has been neutralised. At this point, sulphate concentrations remain constant, and aerosol water content increases with increasing nitrate. In addition to these constraints, the ex-

isting aerosol will only take up water if the relative humidity is sufficiently high (i.e. greater than the DRH) (Seinfeld and Pandis, 2006).

During the 28 day simulation, the mean aerosol water content in the AA80 run is 12.1% less than in the high resolution FRA10 run; this difference reaches up to 36% less in some regions (Figure 4). We explore the aerosol and gaseous species within the equilibrium system in Figure 5, which

shows the mean percent difference of the total column amounts of sulphate, nitrate, ammonia, and nitric acid between the FRA10 and AA80 simulations. Overall, the changes are small in the column amounts of the various species with average percent differences of +4.7%, -2.6%, -6.6%, and +6.1% for sulphate, nitrate, ammonia and nitric acid, respectively. Ammonia and nitrate are both slightly underestimated in the AA80 run; however, the magnitude and spatial distribution of the differences



do not match the underestimation in aerosol water content. This is due to the fact that the aerosol
species do not take up water under all conditions (as discussed above), and so looking at mean
column differences over the full duration of the simulation may miss important information.

In MARS, four main regimes are defined as follows: High RH and High fraction of Ammonia to
sulphate (HRHA); High RH and Low Ammonia (HRLA); Low RH and High Ammonia (LRHA);

and Low RH and Low Ammonia (LRLA). High RH refers to a humidity greater than or equal to
40%, whereas a low RH is less than 40%. A value of 40% was used to approximate the RH of crys-
tallisation of ammonium nitrate and ammonium sulphate. A high fraction of ammonia to sulphate
refers to a fraction greater than or equal to 2.0, whereas a low fraction is less than 1.0. The model
includes regimes for mass fractions between 1.0 and 2.0; however, they are not included in this anal-

ysis due to their relatively infrequent occurrence during the simulation. Because nitrate can only
exist once there is sufficient ammonia to neutralise sulphate and can only absorb water at relative
humidities above its DRH, the HRHA regime is the only regime in which nitrate can uptake water.

The amounts of each chemical species and the total water content within each of the regimes are
compared in Table 3. LRHA is not included in the table because the aerosol water content is set to

zero in this regime. This is due to the fact that although there may be sulphate and nitrate present,
there is insufficient humidity to transition them to their aqueous states. One point to note regarding
the LRHA regime, however, is that the AA80 simulation spends approximately 12% of its time in
this regime, compared to 8% for the FRA10. This may therefore be a small contributing factor to the
underestimation in aerosol water in AA80.

Looking at the overall differences, we see similar behaviour as Figure 5 there is a large decrease
in aerosol water, with small changes in all other species. By exploring the different regimes, we
can see that the chemical system spends most of its time within the HRHA and the LRLA regimes.
We also see that the average aerosol water content is lower in AA80 compared to FRA10 in all
three regimes; however, the absolute values of the concentrations in the HRHA regime are orders

of magnitude higher than in the other two regimes, indicating that the HRHA has the largest impact
on total aerosol water content. This is the high-humidity, ammonia-rich regime described above. In
this regime both sulphate and nitrate aerosol can uptake water; this is the *only* regime in WRF-Chem
in which nitrate aerosol can contribute to the total aerosol water content. In the HRHA regime, the
AA80 simulation underestimates both sulphate and nitrate aerosol, however, the underestimation in

nitrate is roughly 2 orders of magnitude larger than that of sulphate. Also note that even though there
is less ammonia in AA80, this does not impact the amount of time the system spends within the
HRHA regime, meaning there is enough ammonia present to fully neutralise sulphate, but there is
less leftover to form nitrate within the HRHA regime. Thus, although there is a small decrease in
nitrate overall, the decrease is much larger under the conditions that are most favourable for nitrate

to take up water. This leads to less aerosol water in the AA80 run.





The vertical profiles of ammonia, accumulation mode nitrate, accumulation mode aerosol water content, and extinction from the FRA10 and AA80 simulations are shown in Figure 6. The vertical profiles of ammonia (Figure 6a) reveal a ∼30% underestimation at the surface in the AA80 simulation, with very little differences at higher altitudes. The vertical profiles of nitrate (Figure 6b),

however, show differences in the vertical distribution of nitrate at altitudes up to 9 km with the AA80 simulation having more nitrate at the surface, significantly less nitrate in the boundary layer (BL) and more nitrate above the BL compared to the FRA10 simulation. While the difference in total nitrate concentration between the two simulations is small (less than 3%), the differences in the BL reach up to 20%. The boundary layer is characterised as having high relative humidity and lower

temperatures than the surface, which are the conditions under which nitrate most readily absorbs water. It is therefore this underestimation in BL nitrate that leads to an underestimation in aerosol water content (Figure 6c), and, ultimately, extinction (Figure 6d). Aerosol water content is largely unaffected by the small increases in nitrate at the surface and above the BL because nitrate does not efficiently take up water under these conditions.

Although previous studies showed the importance of sub-grid RH variability (Haywood et al., 1997; Bian et al., 2009), in our case compositional variability is more important. This is easily shown by doing an experiment where only RH is averaged and not the aerosol. In that case, AOD is only 8.7% lower than in AA10 (rather than 13.1% lower when aerosol composition is varied).

Understanding the mechanism causing the underestimation of BL nitrate in the low resolution sim-
ulation is complicated by the fact that nitrate is part of a coupled equilibrium system. The question remains: what factors contribute to the simulated changes in nitrate? While the complete explanation for the changes in nitrate is difficult to constrain unambiguously, the following sections explore a number of mechanisms that may contribute to these changes.

### 3.1.2 Investigating changes in nitrate: Impact of equilibrium system

In a previous study, Metzger et al. (2002) coupled a gas-aerosol equilibrium scheme to a global atmospheric chemistry-transport model and tested the effect of decreasing the full model resolution from 10°x 7.5° to 2.5°x 2.5° on aerosol nitrate. They found that boundary layer nitrate concentrations were 30-80% lower in the low resolution run. They attributed these large differences to the fact that aerosol nitrate formation non-linearly depends on the concentrations of its precursor gases.

To test whether the changes in boundary layer nitrate concentrations in the current study are related to changes in resolution of aerosol and gaseous species within the equilibrium system, we conduct an alternative AA80 simulation during which all aerosols and gases are averaged over the lower resolution grid *except* the species involved in the equilibrium, namely, sulphate aerosol, ammonium aerosol, ammonia, nitric acid, and nitrate aerosol. The results from this simulation show that the

differences in aerosol water content between the FRA10 and the altered AA80 simulations virtually disappear (0.1% difference), confirming that the underestimation in aerosol water content in the





AA80 simulation can be attributed to neglecting the sub-grid variability of species within the nitrate equilibrium system.

We also perform a number of sensitivity simulations using a box model version of the aerosol water equilibrium system. The box model is identical to the coded version within WRF-Chem and simulates the gas-aerosol partitioning and subsequent aerosol water uptake. The box model requires as input the initial concentrations of the five species involved in the equilibrium, temperature, and relative humidity and produces as output the equilibrium concentrations of each species as well as the aerosol water content at equilibrium. In the sensitivity tests, the input concentrations of four of the aerosol/gaseous species, as well as the temperature and relative humidity, are held constant and the input concentration of the fifth species is randomly sampled from a lognormal distribution. The standard deviation of this lognormal distribution characterises the spatial variability of the input aerosol concentration. A high standard deviation corresponds to a high spatial variability, thereby mimicking a high model resolution. The sensitivity tests compare the difference in equilibrium concentrations when the input concentration of one aerosol/gaseous species has a high degree of variability versus a low degree of variability. The test is therefore analogous to measuring the response of the equilibrium system to a decrease in resolution of one aerosol/gaseous species while holding all other parameters constant.

Each test case consists of 1000 random samples; the high variability case has a standard deviation approximately 1.5 times greater than the low variability case, which matches the values of the standard deviations calculated from the WRF-Chem model output. The mean concentrations of the high variability and low variability lognormal distributions are identical and set to the mean concentration of that particular gas or aerosol from the FRA10 WRF-Chem simulation. The input concentrations of the other four aerosol and gaseous species remain constant and are set to their corresponding mean values in the FRA10 WRF-Chem simulation. The sensitivity tests are performed using mean concentrations at two different levels: model level 0 (0 km) and model level 6 ($\sim$1 km).

We conduct the tests at six different relative humidities (0.50, 0.60, 0.70, 0.80, 0.90, 0.95) and four different temperatures (275K, 280K, 285K, 290K). We change the input variability of each of the five aerosol/gaseous species from high to low one at a time, so that a total of 120 sensitivity tests are performed at each model level (5 aerosol/gaseous species x 6 relative humidities x 4 temperatures).

An example of the results from one sensitivity test is shown in Figure 7. In this particular test, the input concentrations of ammonia are randomly sampled at a high variability (in blue) and at a low variability (in red). This plot highlights the non-linear relationship between many of the species and ammonia. Remember that the means of the high and low variability input ammonia distributions are identical, so that if the relationships were linear, there would be no difference in the mean equilibrium concentrations. However, we can see that the mean concentration of nitric acid is lower in the low variability run, whereas the mean concentrations of nitrate, ammonium, and aerosol water are higher in the low variability run. There is also a small decrease in the mean equilibrium concentra-



tion of ammonia when the distribution of its input concentrations has a lower variability. The mean
equilibrium concentrations of sulphate are unaffected by ammonia variability.

The results from all of the sensitivity tests are summarised in Figure 8 using surface concentrations
(left column) and boundary layer concentrations (right column) as inputs to the equilibrium calcu-
lations. The first row shows the effect of reducing the variability of each aerosol/gaseous species
on ammonia. The y-axis represents the percent difference in the mean equilibrium concentrations
of ammonia between the low variability and high variability runs (low - high). Each colour repre-
sents a different species whose variability was altered, e.g. the blue dots represent the runs when the
variability of input sulphate was reduced. Each different dot within the same colour represents a test
performed at a unique relative humidity and temperature value with darker colours corresponding
to higher relative humidities and larger dots corresponding to higher temperatures. The second row
shows the same for nitrate, and the third row shows the same for aerosol water.

Figure 8a shows that reducing the variability of nitrate, ammonia, and ammonium all result in
lower ammonia concentrations at the surface by 10-15%. In the boundary layer, we see much higher
percent differences, which is a consequence of lower ammonia concentrations having a higher sensi-
tivity to changes in aerosol and gas variability. Reducing sulphate variability again produces mixed
responses in mean ammonia equilibrium concentrations, and the rest of the aerosol/gaseous species
result in lower ammonia concentrations by up to 30%.

Looking at the impact of aerosol and gas variability on nitrate (Figure 8b), we see the opposite
trend as ammonia. At the surface, reducing the variability of nitrate, ammonia, and ammonium re-
sults in higher mean nitrate concentrations up to 20%. While most of the changes result in higher
nitrate concentrations, we see decreases in nitrate of close to 10% when the variability of sulphate,
nitric acid, and nitrate is reduced at the lowest relative humidity and highest temperature. The bound-
ary layer shows a much more variable picture. While there is no strict trend, we tend to see less nitrate
in the low variability run at lower relative humidities and higher temperatures. Aerosol water content
(third row) follows a similar trend to nitrate except that the percent differences in the boundary layer
are smaller in magnitude.

Relating these sensitivity test results back to the simulated changes in the WRF-Chem, there are a
few key observations to note. Firstly, these tests highlight the complicated nature of this equilibrium
system. By simply changing the degree of variability of one input parameter, large differences arise
in equilibrium concentrations of all aerosol and gaseous species (expect sulphate) within the equilib-
rium. In an additional test during which we changed the degree of variability of *two* input parameters
(not shown), the relationships become significantly more scattered and less predictable. In the WRF-
Chem simulations, the variability of all aerosol and gaseous species are changed simultaneously,
which makes the subsequent impact on the equilibrium system difficult to predict.

Nevertheless, the sensitivity tests provide significant insight. The differences in the some of the
sensitivity runs are of similar magnitude to the differences between the FRA10 and AA80 simula-





tions. The majority of the sensitivity tests show that lower aerosol and gas variability results in less ammonia and more nitrate at the surface, which follows the trend observed in the FRA10 and AA80 WRF-Chem simulations. While the impact on nitrate in the boundary layer is more variable, we do see reductions in mean nitrate concentrations, particularly at lower relative humidities and higher

temperatures. Aerosol water shows a smaller negative effect, likely due to the fact that the largest reductions in nitrate occur under unfavourable conditions for water uptake.

### 3.1.3 Investigating changes in nitrate: Impact of convective transport

In aerosol simulations, nitrate-containing air in the boundary layer mixes with layers above and below. In the high resolution run, the mixing occurs as normal, with some nitrate-containing air be-

ing removed from the BL by mixing with adjacent layers. When nitrate concentrations are low or depleted in the high resolution run, further removal of nitrate can only occur after it has been replenished by advection or emission/secondary production. In the AA80 run, the removal mechanism occurs at a high resolution but nitrate concentrations are spread over the low resolution grid box. In this scenario, the nitrate concentrations are continuously averaged and re-distributed over the large

grid area so that the nitrate that has been removed from the BL by the high-resolution mixing is instantaneously replenished by the averaging over neighbouring grid boxes. It is therefore possible that more nitrate is being depleted from the BL in the low resolution run due to the continuous spreading of nitrate over areas where it has already been removed by convective transport.

We repeated the FRA10 and AA80 simulations but with convective transport turned off. Figure

9 demonstrates the effect of turning off convection on the vertical profiles of ammonia (a) and accumulation mode nitrate (b). Ammonia shows very little difference from the original FRA10 and AA80 simulations, i.e. the underestimation of ammonia at the surface in AA80 persists when convective transport is turned off. On the other hand, the underestimation of nitrate in the BL in the original AA80 simulation disappears when convective transport is turned off and results in a higher

overestimation at the surface. This agrees with results from the sensitivity tests in the previous section, which showed a tendency to simulate less ammonia and more nitrate at the surface at lower resolutions. Also, with the disappearance of the underestimation of BL nitrate, we no longer see an underestimation in the column amount of accumulation mode aerosol water (not shown).

At first glance, this appears to explain the differences in aerosol water content between the FRA10

and AA80 simulations. However, further investigation reveals a more complicated picture. Our results show that although convective transport likely plays a role in the underestimation of nitrate in the BL, it does not explain the full story. To explore the impact of convective transport in more detail, we focus on a 5-day period from May 3 - 7 during which there was a large convective rainfall event confined to one side of the domain (see the top panel in Figure 10). Figure 11 shows the differences

in column aerosol water content between FRA10 and AA80 for this period with convective transport turned on (a) and turned off (b). In the original AA80 simulation, one sees the underestimation in





aerosol water content, this time confined to the lefthand side of the domain. This is to be expected as the relative humidity is much higher on this side of the domain during this time period, which also contributes to the large convective rainfall event. Once again, when convective transport is turned

off, the underestimation in aerosol water largely disappears.

However, if one examines the two sides of the domain separately, we see a different trend. Figure 10 shows the mean vertical profile of accumulation mode nitrate, split up by area — the lefthand side where there is significant convection, and the righthand side where there is no convection. The top row shows the spatial distribution of the cumulative convective rainfall from May 3 - 7; the

second row shows the vertical profiles of nitrate when convective transport is left on; and the third panel shows the same vertical profiles for the simulations where convective transport is turned off. The middle panels show that nitrate is underestimated in the BL in the AA80 simulation on both sides of the domain, even though there is very little convection on the righthand side. Thus when convective transport is turned off, one sees the underestimation in nitrate disappear on the side of the

domain where there is significant convective transport; however, the underestimation persists on the righthand side of the domain, where there is no convection.

It is likely that a combination of convective effects, which tend to cause underestimations in BL nitrate under conditions of high relative humidity, and non-linearities in the equilibrium system, which tend to cause decreases in BL nitrate under conditions of low relative humidity, lead to the

differences in the FRA10 and AA80 simulations.

### 3.2 Effect of aerosol sub-grid variability on CCN

Figure 12 presents results of simulated CCN at 0.5% supersaturation for the FRA10, AA40, AA80, and AA160 simulations. At lower resolutions, the simulated monthly averaged CCN is overestimated in all regions. Compared to FRA10, the overestimation of CCN increases from an average of 17.8%,

to 27.3% to 36.0% as the aerosol resolution is decreased to 40 km, 80 km, and 160 km, respectively.

Figure 13 shows the mean spatial distribution of accumulation mode number concentration for the FRA10 simulation coarsened to the low resolution grid, the AA80 simulation, and the percent difference between them. The AA80 accumulation mode number concentration is also significantly overestimated compared to the high resolution run by an average of 27.4%. We can readily see that

the overestimation in CCN is nearly equivalent in magnitude and spatial distribution to the overestimation in accumulation mode number concentration, indicating that changes in CCN at this supersaturation are dominated by changes in accumulation mode number. Pattern correlation analysis confirms that differences in CCN and accumulation mode number are indeed highly correlated with an average correlation of 0.99. There are also small overestimations in mean Aitken mode (+10%)

and coarse mode (+3%) number concentrations in AA80; however, it is clear that accumulation mode number is the dominant contributor to CCN under these conditions. When the supersaturation is in-





creased to 3%, the overestimation in CCN decreases to 17.2%, indicating a larger contribution from Aitken mode aerosols where the discrepancies between the high and low resolution runs are smaller.

The marked increase in CCN and accumulation mode number concentration is also apparent in
their vertical profiles, shown in Figures 14a and b. The increase in both CCN and accumulation mode number concentration exists at all altitudes from 0 to 12 km, with the largest increase at the surface.

At altitudes above 2 km, the overestimation of aerosol number is due to the averaging of gaseous concentrations within the model. We perform an alternative AA80 simulation where only the aerosol fields are averaged over a lower resolution grid, and the gaseous fields remain on the original high
resolution grid. In this case, the differences in accumulation mode number concentration at altitudes above the surface largely disappear. Figure 15 shows the vertical profile of accumulation mode number concentration from FRA10 and the alternative AA80 simulation with high resolution gas fields. One can see that the overestimation in the original AA80 run at altitudes greater than 2 km is significantly reduced. Accounting for the overestimation in the rate of nucleation, the overall bias reduces
from +27.3% to +10.3%.

To explain this behaviour we look at the rate of new particle production by nucleation. In the AA80 simulation, the nucleation rate is 25% higher in the upper troposphere than in the FRA10 simulation. A higher nucleation rate results in a higher concentration of Aitken mode particles in the upper troposphere, which leads to higher accumulation mode number concentration as there are
more particles available to grow into the larger mode. These results are highlighted in Figures 14c and d, which show the increase in nucleation rate and the corresponding increase in Aitken mode number concentrations above the surface, particularly between 6 and 9 km where the difference in nucleation rate is greatest.

The standard WRF-Chem nucleation scheme was used in these simulations. This scheme is a
simple parameterisation of homogeneous nucleation in the sulphuric acid-water system (Kulmala et al., 1998). Within this parameterisation, the nucleation rate depends non-linearly on temperature, relative humidity, and sulphuric acid vapour concentration. Because the meteorological parameters are identical in the full resolution and the aerosol averaged simulations, the non-linear dependence of the nucleation rate on sulphuric acid vapour concentration must be the source of the discrepancy
between the FRA10 and AA80 simulations.

The concentration of sulphuric acid vapour is determined by its chemical production and loss due to nucleation and condensation. Sulphuric acid vapour is produced by the reaction of the hydroxyl radical (OH) and sulphur dioxide gas ($SO_2$). Inspection of the changes in sulphuric acid vapour concentration between the FRA10 and AA80 runs shows very little difference; however, the concen-
tration of OH is overestimated in the AA80 simulation by 15 - 20% in the upper troposphere (shown in Figure 16). Even though there is very little difference in overall concentration of sulphuric acid between the two runs, the overestimation in OH leads to an increased rate of oxidation of sulphur dioxide, causing an increase in the chemical production tendency of sulphuric acid by 26.8% at high



altitudes in the AA80 run). The excess sulphuric acid produced is then subsequently used for new
particle production and condensational growth in the upper troposphere, resulting in high aerosol
number concentrations at these altitudes.

Although OH chemistry in the upper troposphere involves a myriad of complex reactions, the
concentration of OH has been found to largely depend on its primary production rate from ozone
photolysis (Jaeglé et al., 2001). Ozone production is known to be dependent on its precursor concen-
trations in a non-linear manner, particularly $NO_x$ (NO + $NO_2$). Previous work has shown that ozone
production is relatively inefficient at high concentrations of $NO_x$ found in near-source areas com-
pared with low concentrations typical of remote regions (Sillman et al., 1990). This non-linearity can
therefore have a large impact on model-simulated ozone concentrations due to the artificial mixing
of its precursor gases over large grid areas, resulting in excessive production of ozone and, conse-
quently, higher hydroxide concentrations (Esler et al., 2004). In the AA80 simulation, the artificial
mixing of aerosols and trace gases is likely the cause of the higher rate of ozone production, which
is up to 3.5 times greater than in the FRA10 simulation.

Convective transport also plays a role in the overestimation due to nucleation. This mechanism
lofts gaseous species to altitudes above the boundary layer where ozone and hydroxide production
is more efficient and where their lifetimes are longer. Turning off convective transport of aerosol and
gaseous species produces a similar result to the altered, no-gas-averaged AA80 simulation, reducing
the overestimation of CCN from +27.3% to +10.7% (not shown).

Several previous studies have shown that ozone production is overestimated at lower model reso-
lutions (e.g. Esler et al., 2004; Wild and Prather, 2006). This is due to the non-linear dependence of
ozone production on its gaseous precursor concentrations, with most of the production occurring on
short time scales close to regions with high precursor emissions. Artificially diluting ozone and its
precursor gases over a model grid box effectively increases the time scale over which its chemical
production occurs. Also, artificial dilution of gas fields acts to exaggerate the importance of con-
vection, enhancing the export of longer-lived gases to the mid- and upper troposphere where ozone
production is more efficient.

In summary, mixing of aerosol and gaseous fields over an 80 km grid results in an increase in
ozone production, which is lofted to higher altitudes and leads to higher concentrations of OH in the
upper troposphere. Enhanced OH concentrations result in faster oxidation of $SO_2$, producing higher
concentrations of sulphuric acid, which promotes the formation of new aerosol particles in the upper
troposphere. Higher number concentrations at altitudes above 2 km lead to increased CCN. This
mechanism accounts for a significant portion of the total bias in CCN.

The overestimation in accumulation mode number at the surface is related to dry deposition pro-
cesses. When both nucleation and aerosol/gas dry deposition processes are turned off, the difference
between FRA10 and AA80 virtually disappears. The likely mechanism behind the overestimation
due to dry deposition is that by simulating aerosols over a lower resolution grid than the underlying



terrain in WRF-Chem, aerosols in coastal regions that are normally deposited over land are being spread over the ocean where the deposition velocities are set to zero, causing a build up of aerosol over oceans and other bodies of water. Examining the spatial distribution of the differences in accumulation mode number at the surface shows a strong overestimation over the ocean areas and the

English Channel. The nature of this particular domain may amplify this affect due to the extensive coastal regions within the domain. The magnitude of this dry depositional effect on a global scale is unclear and requires further investigation.

### 3.3    Full resolution comparisons

As discussed in the introduction, a common method for investigating the impact of sub-grid variabil-

ity on model predictions of the aerosol effect on climate is to vary a model's resolution and analyse the resulting effect this has on aerosol fields. While this method can provide some insight into the differences in model behaviour at different grid spacings, it is limited in its ability to pinpoint the processes that contribute to these differences.

We highlight this difficulty by comparing results from the FRA10 and FRA80 simulations, where

the full resolution of the model has been changed from 10 km to 80 km. Figures 17 and 18 show mean AOD and CCN fields for each of these runs, respectively.

Figure 17a shows AOD at 10 km resolution and Figure 17c at 80 km. The AOD fields from the higher resolution run are coarsened to the low resolution grid (Figure 17b) before taking the percent difference between the two runs (Figure 17d). The changes in AOD due to varying the full model

resolution are drastically different from the changes in AOD due to varying the resolution of the aerosols only (Figure 3).

Decreasing the model resolution from 10 km to 80 km results in a 20 – 40% underestimation of AOD over the English channel region, and a 20% overestimation in AOD in the northern regions of the domain. Further investigation reveals that the differences in AOD are again linked to changes

in aerosol water content; however, the underlying mechanisms causing the changes in aerosol water are much less clear. Not only are there changes in aerosol composition, as seen in the "aerosol averaged" comparisons, there are also large changes in average daily relative humidity and temperature and other meteorological parameters, which further complicate the gas-aerosol thermodynamic equilibrium.

In fact, the amount of convective rainfall is more than 50% less in the FRA80 simulation compared to FRA10. Since wet deposition and convective transport are important aerosol removal mechanisms this underestimation in rainfall likely masks many of the changes we observed due to aerosol variability.

The changes in CCN due to varying the full model resolution are also starkly different from the

changes due to varying aerosol resolution only (Figure 18). Whereas CCN was largely overestimated in the AA80 simulation, CCN is now significantly underestimated (on average -33.0%) in all





regions of the FRA80 domain. While this underestimation is also linked to changes in accumulation mode number concentration as seen in the "aerosol averaged" simulations, the FRA80 simulation shows an underestimation in the nucleation rate of -24.3% at its peak. In the FRA80, convection is significantly weakened. While a decrease in rainfall could act to increase aerosol concentrations, a weakening of convective transport could significantly affect gas chemistry in the upper troposphere, thereby altering the nucleation rate and secondary formation of aerosols. These competing interactions, along with other changes in meteorology, make it difficult to gain an understanding of the processes governing the simulated decrease in CCN in FRA80.

Additionally, while we were able to offer possible explanations to the changes seen in AOD and CCN in the FRA80 simulation, the insight to these changes came from the previous analysis of the AA80 simulation, further highlighting the usefulness of isolating the effect of aerosol variability.

## 4 Conclusions

This study investigates the impact of subgrid variability, neglected in global microphysical aerosol models, on two important aerosol properties: aerosol optical depth and cloud condensation nuclei, which serve as proxies for aerosol-radiation and aerosol-cloud interactions, respectively. It introduces a novel technique to isolate the effect of aerosol variability from other sources of model variability by varying the resolution of aerosol and trace gas fields while maintaining a constant resolution in the rest of the model. The aerosol resolution is varied to 40 km, 80 km, and 160 km (AA40, AA80, and AA160) and compared to a baseline high resolution run at 10 km (FRA10). The simlulations are run for a month-long period in May 2008.

Decreasing the resolution of the aerosol fields results in an underestimation of monthly averaged AOD by 10% (for AA40) to 16% (for AA160) over the whole domain, with some regions showing decreases of up to 30% in AA160. Decreasing the resolution of aerosol and gaseous fields results in an overestimation of CCN by an average of 18% (for AA40) to 36% (for AA160) over the entire domain.

The changes in AOD are linked to changes in accumulation mode aerosol water content, which is determined by the sulphate-nitrate-ammonium gas/particle partitioning equilibrium. In the AA80 simulation, nitrate aerosol concentrations in the boundary layer are approximately 20% less compared to the FRA10. Water uptake by nitrate is most efficient in the boundary layer, where relative humidity is high and temperature is low relative to the surface; therefore, this underestimation of nitrate in the aerosol averaged runs leads to an underestimation of aerosol water. Box model tests of the nitrate equilibrium system demonstrate that neglecting variability of aerosol and gaseous species within the system has a highly non-linear effect on equilibrium concentrations. The underestimation of nitrate in the boundary layer is likely due to a combination of the response of the non-linear



equilibrium system to changes in aerosol and gaseous variability and of convective transport, which
removes more nitrate in the low resolution run.

Over the past decade, GCMs have been incorporating nitrate aerosol in direct radiative forcing
calculations. In the AeroCom Phase II direct radiative forcing study, eight of the sixteen models
currently use an equilibrium parameterisation for nitrate and aerosol water uptake, and two more are
in the process of incorporating them into their models (Myhre et al., 2013). The results presented in
this paper indicate that accurate representation of aerosol radiative effects requires a realistic model
of water uptake by aerosols, including sub-grid spatial variation in aerosol chemical composition.

While the variability in relative humidity is certainly an important factor in determining aerosol
radiative forcing, we show that even when using identical resolution relative humidity, AOD is still
underestimated at GCM resolutions. These results suggest that at least some of this underestimation
in AOD in previous studies is due to the impact of sub-grid variability of aerosol composition on
water uptake as well as variability in RH. Similar results have been shown with modelling studies
over the Netherlands, whose environment is characterised by its high concentrations of ammonia
and nitric acid due to agricultural activity (Roelofs et al., 2010; Derksen et al., 2011).

The changes in CCN are linked to changes in accumulation mode number concentration, which
is also overestimated by a similar degree. At the surface, the overestimation of CCN is related to
differences in dry deposition processes over land and ocean when averaging aerosols over a lower
resolution than the underlying terrain. At higher altitudes, the increase in accumulation mode number
is influenced by enhanced trace gas chemistry. The artificial dilution of trace gases results in an
increase in the production of ozone, leading to increased OH, which results in higher concentrations
of sulphuric acid vapour. With more sulphuric acid vapour available for nucleation, the number
of Aitken mode particles increases at high altitudes, leading to an increase in accumulation mode
number. Convective transport again plays an important role in the simulated differences in CCN by
lofting the trace gases into the troposphere where the gases have longer lifetimes and their reactions
are more efficient.

We know from previous research that mixing of gases at global model grid-scales can result in
large discrepancies in simulated and observed gaseous concentrations. And while it has been well-
documented that gas-phase chemistry is dependent on model resolution, this study demonstrates that
these gas-phase discrepancies can have a significant impact on aerosol properties through secondary
aerosol formation.

Comparisons between the full resolution run at 10 km and the full resolution run at 80 km highlight
the difficulty in identifying the mechanisms that cause differences in aerosol properties at different
model resolutions. The changes in AOD and CCN between these two runs are different in both sign
and magnitude from the changes in AOD and CCN in the "aerosol averaged" runs where only the
aerosol resolution is varied. In these comparisons, it is not feasible to determine if discrepancies
between the high and low resolution simulations are due to neglecting sub-grid variability of me-


teorological, dynamical, or aerosol processes. Large differences in meteorological parameters such as convective rainfall and relative humidity could be masking effects caused by neglecting aerosol

variability.

This paper demonstrates that aerosol variability existing at sub-grid scales can have a significant impact on important aerosol properties, such as AOD and CCN. Processes most affected by neglecting aerosol sub-grid variability are gas-phase chemistry and aerosol uptake of water through the aerosol/gas equilibrium reactions. The inherent non-linearities in these processes result in large

changes when aerosol and gaseous species are artificially mixed over large spatial scales, as is the case in the current generation of global microphysical aerosol models. These changes in aerosol and gas concentrations are exaggerated by convective transport, which transport these altered concentrations to altitudes where their effect is more pronounced. Future aerosol model development should focus on accounting for the effect of sub-grid variability on these processes at global scales in order

to more accurately predict the aerosol effect on climate.

*Acknowledgements.* This work was supported by the Natural Sciences and Engineering Research Council of Canada and the Clarendon Fund. The research leading to these results has received funding from the European Research Council under the European Union's Seventh Framework Programme (FP7/2007-2013)/ERC grant agreement no. FP7-280025 and the UK National Environment Research Council project GASSP (NE/J024252/1).



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





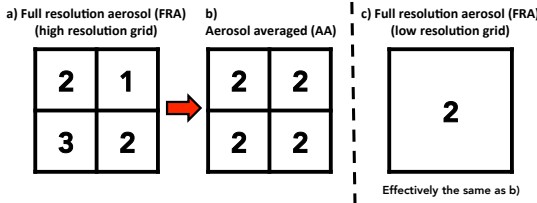

Figure 1: Conceptual description of experimental design. The first panel represents high resolution aerosol fields (a). In the second panel, the mean value of these fields is assigned to each of the high resolution grid points (b), which is effectively the same as a lower resolution grid with the same value (c).

Sillman, S., Logan, J. A., and Wofsy, S. C.: A regional scale model for ozone in the United States with subgrid representation of urban and power plant plumes, Journal of Geophysical Research: Atmospheres, 95, 5731–5748, doi:10.1029/JD095iD05p05731, 1990.

Skamarock, W. C. and Klemp, J. B.: A time-split nonhydrostatic atmospheric model for weather research and forecasting applications, Journal of Computational Physics, 227, 3465–3485, doi:10.1016/j.jcp.2007.01.037, 2008.

Stockwell, W., Middleton, P., Chang, J., and Tang, X.: The 2nd Generation Regional Acid Deposition Model Chemical Mechanism for Regional Air-Quality Modeling, Journal of Geophysical Research-Atmospheres, 95, 16 343–16 367, doi:10.1029/JD095iD10p16343, 1990.

Twomey, S.: Pollution and Planetary Albedo, Atmospheric Environment, 8, 1251–1256, doi:10.1016/0004-6981(74)90004-3, 1974.

van der Gon, H. D., Kuenen, J., and Butler, T.: A Base Year (2005) MEGAPOLI Global Gridded Emission Inventory (1st Version). Deliverable D1., MEGAPOLI Scientific Report 10-13, 2010.

Wainwright, C. D., Pierce, J. R., Liggio, J., Strawbridge, K. B., Macdonald, A. M., and Leaitch, R. W.: The effect of model spatial resolution on Secondary Organic Aerosol predictions: a case study at Whistler, BC, Canada, Atmospheric Chemistry and Physics, 12, 10 911–10 923, doi:10.5194/acp-12-10911-2012, 2012.

Weigum, N. M., Stier, P., Schwarz, J. P., Fahey, D. W., and Spackman, J. R.: Scales of variability of black carbon plumes over the Pacific Ocean, Geophysical Research Letters, 39, 15 804, doi:10.1029/2012GL052127, 2012.

Wesely, M.: Parameterization of Surface Resistances to Gaseous Dry Deposition in Regional-Scale Numerical-Models, Atmospheric Environment, 23, 1293–1304, doi:10.1016/0004-6981(89)90153-4, 1989.

Wild, O. and Prather, M. J.: Global tropospheric ozone modeling: Quantifying errors due to grid resolution, Journal of Geophysical Research: Atmospheres, 111, doi:10.1029/2005JD006605, 2006.

Wild, O., Zhu, X., and Prather, M.: Fast-j: Accurate simulation of in- and below-cloud photolysis in tropospheric chemical models, Journal of Atmospheric Chemistry, 37, 245–282, doi:10.1023/A:1006415919030, 2000.

Williamson, D.: Convergence of atmospheric simulations with increasing horizontal resolution and fixed forcing scales, Tellus A, 51, 663–673, 1999.





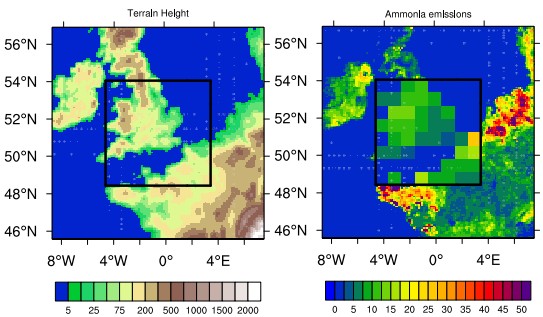

Figure 2: First panel shows terrain height (in metres) of the WRF-Chem domain. The outer frame represents the total high resolution 10 km domain; the inner box represents the region over which the averaging technique is applied. The second panel shows daily averaged ammonia emissions in mol km$^{-2}$ h$^{-1}$. The inner domain has been averaged over 80 km.

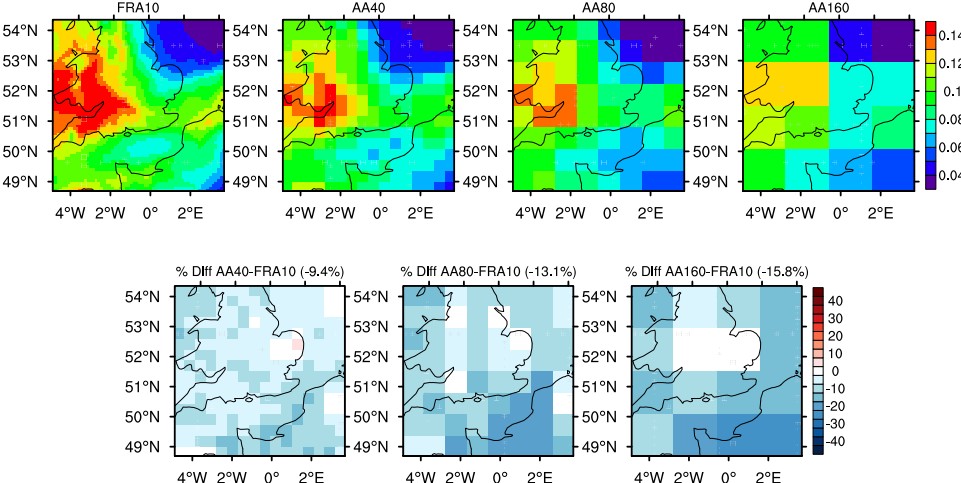

Figure 3: Simulated mean spatial distribution of AOD for the FRA10, AA40, AA80, and AA160 runs (top row) from May 3 - 31, 2008. The bottom row represents the percent difference between each aerosol averaged run and FRA10. The number in brackets in the bottom row represents the mean percentage difference in the two runs.





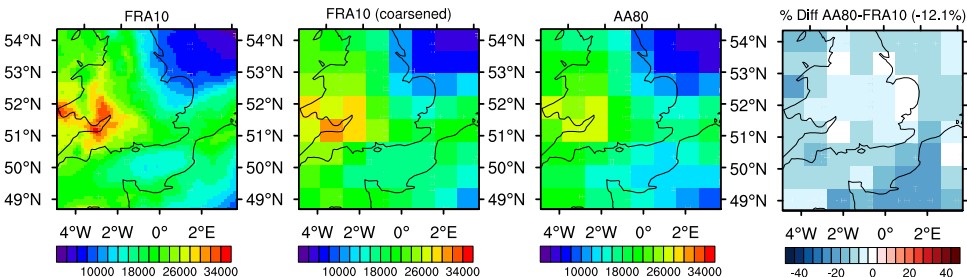

Figure 4: Simulated mean spatial distribution of column amount accumulation mode aerosol water content in $\mu$g m$^{-2}$ for the FRA10 simulation, the FRA10 simulation coarsened to the 80 km grid, the AA80 simulation, and the percent difference between the two. The number in brackets on the third panel represents the mean percentage difference in the two runs.

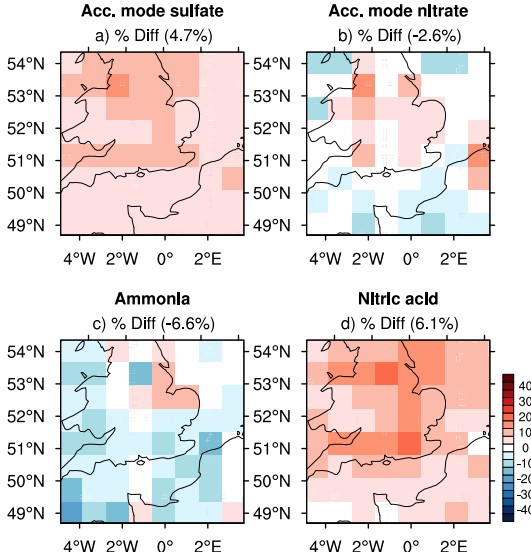

Figure 5: Percent differences between the AA80 and FRA10 (AA80 - FRA10) simulations in the mean spatial distribution of various species in the aerosol water equilibrium system.





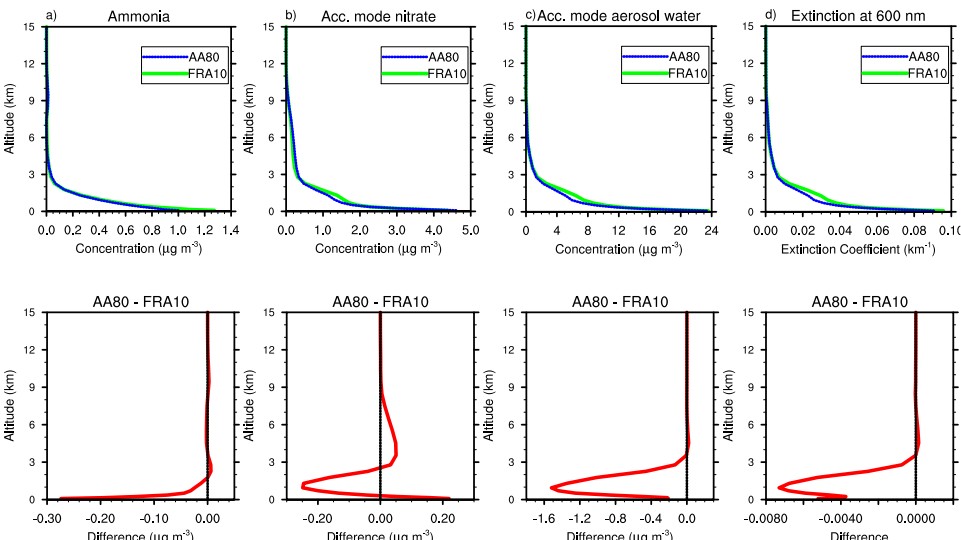

Figure 6: Vertical profiles of selected species in the gas-aerosol equilibrium system in $\mu$g m$^{-3}$ for the FRA10 and AA80 simulations (left column) and the absolute differences between the two simulations (AA80-FRA10, right column). The species include ammonia, (a), accumulation mode nitrate (b), accumulation mode aerosol water content (c), and extinction (d).

Table 1: Description of WRF-Chem simulations analysed in this study.

| Abbreviation | Simulation description |
|---|---|
| FRA10 | Entire model is run at 10 km resolution |
| FRA80 | Entire model run at 80 km resolution |
| AA40 | Model run at 10 km; aerosols and gases at 40 km |
| AA80 | Model run at 10 km; aerosols and gases at 80 km |
| AA160 | Model run at 10 km; aerosols and gases at 160 km |





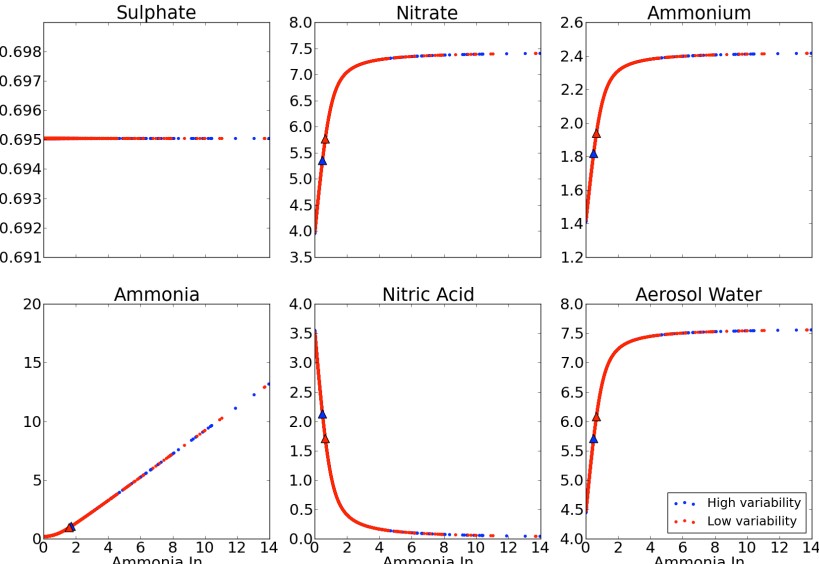

Figure 7: Example of sensitivity test where input ammonia concentrations are randomly sampled at high (blue) and low (red) variability (RH = 0.70, temperature = 280K). The subplots show the equilibrium concentrations (in $\mu g\ m^{-3}$) of individual aerosol/gaseous species as a function of input ammonia concentrations. The blue and red triangles represent the mean equilibrium concentrations of the aerosol/gaseous species in the high and low variability runs.

Table 2: Physical and chemical options used in WRF-Chem configuration.

| Process | WRF-Chem Option | Reference |
|---|---|---|
| Cloud microphysics | Lin | Lin et al. (1983) |
| Long-wave radiation | RRTM | Mlawer et al. (1997) |
| Short-wave radiation | Goddard | Chou and Suarez (1994) |
| Surface layer | Monin-Obukhov | Monin and Obukhov (1954) |
| Land-surface model | Noah LSM | Chen and Dudhia (2001) |
| Boundary Layer scheme | YSU | Hong et al. (2006) |
| Photolysis scheme | Fast-J | Wild et al. (2000) |
| Cumulus parameterization | New Grell (G3) | Grell and Devenyi (2002) |
| Gas-phase mechanism | RADM2 | Stockwell et al. (1990) |
| Aerosol module | MADE/SORGAM | Ackermann et al. (1998); Schell et al. (2001) |



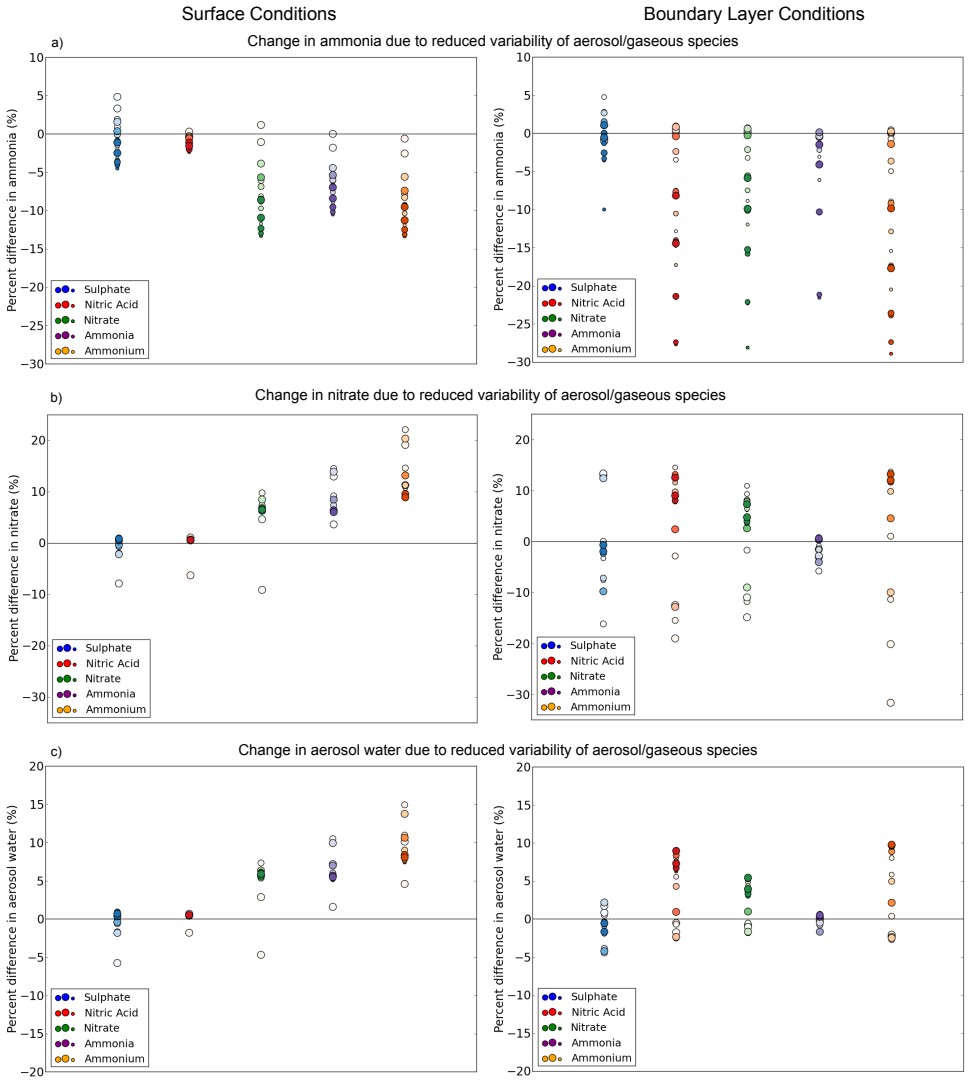

Figure 8: Summarises the results from the box model sensitivity tests for ammonia, nitrate, and aerosol water content. The markers show the difference in the mean ammonia (a), nitrate (b), aerosol water content (c) between the low and high variability runs. Each colour represents a different aerosol/gaseous species whose variability was reduced during the run with darker colours corresponding to higher relative humidities and larger dots corresponding to higher temperatures. The different markers within one colour represent a test performed at a unique relative humidity and temperature value. The left column uses mean surface conditions as input, and the right column shows the same for boundary layer conditions.





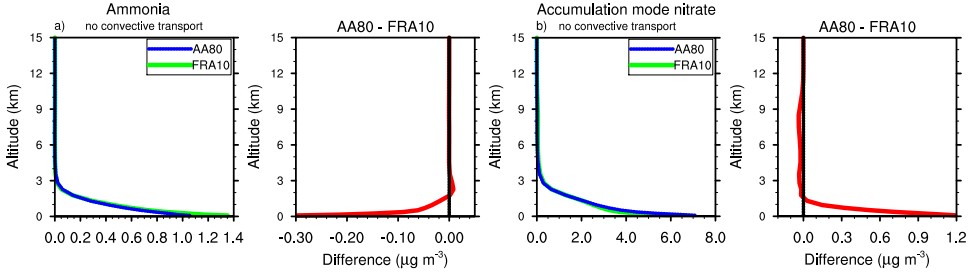

Figure 9: Vertical profiles of ammonia (a) and accumulation mode nitrate (b) concentrations in $\mu$g m$^{-3}$ for the FRA10 and AA80 simulations (left column) and the absolute difference between the two simulations (AA80-FRA10, right column) with convective transport of aerosols and trace gases turned off.

Table 3: Results from regime analysis. The concentrations are averaged over the entire analysis period at all model levels and are expressed in mol m$^{-3}$. The four regimes represent the equilibrium conditions for the formation of sulphate and nitrate aerosol and are divided according to high and low relative humidity (HR, LR), and high and low fraction of ammonia to sulphate (HA, LA). The differences shown correspond to AA80 - FRA10.

|  | % of time | $[SO_4^{2-}]$ | $[NO_3^-]$ | $[NH_3]$ | $[HNO_3]$ | $[H_2O]$ |
|---|---|---|---|---|---|---|
| **Overall** |  |  |  |  |  |  |
| FRA10 | 100 | 2.73 | 13.6 | 13.0 | 20.7 | 245 |
| AA80 | 100 | 2.85 | 13.3 | 11.3 | 21.4 | 226 |
| Difference | - | +0.12 | -0.3 | -1.7 | +0.7 | -19 |
| **HRHA Regime** |  |  |  |  |  |  |
| FRA10 | 40 | 5.19 | 33.6 | 30.7 | 40.5 | 606 |
| AA80 | 44 | 5.07 | 29.5 | 23.4 | 38.1 | 511 |
| Difference | +4 | -0.12 | -4.1 | -7.3 | -2.4 | -95 |
| **HRLA Regime** |  |  |  |  |  |  |
| FRA10 | 11 | 1.54 | 0.016 | 0.032 | 7.28 | 19.9 |
| AA80 | 7 | 1.27 | 0.024 | 0.051 | 6.67 | 12.7 |
| Difference | -4 | -0.27 | +0.008 | +0.019 | -0.61 | -7.2 |
| **LRLA Regime** |  |  |  |  |  |  |
| FRA10 | 41 | 0.66 | 0.004 | 0.032 | 5.58 | 0.92 |
| AA80 | 37 | 0.59 | 0.008 | 0.038 | 5.64 | 0.58 |
| Difference | -4 | -0.07 | -0.004 | +0.006 | +0.06 | -0.34 |

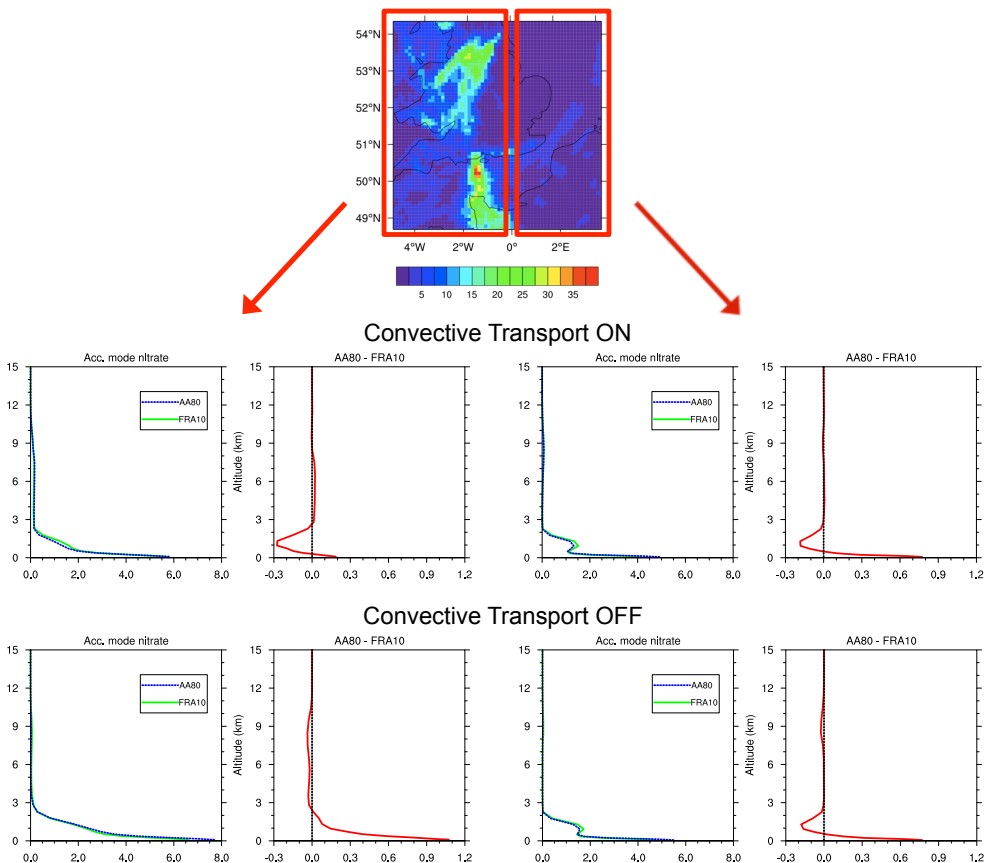

Figure 10: Demonstrates the impact of convective transport on the vertical profile of nitrate. The top panel shows the cumulative convective rainfall from May 3 - 7, 2008 (in mm). The middle panel shows the vertical profile of accumulation mode nitrate (in $\mu$g m$^{-3}$) for the FRA10 (green) and AA80 (blue) and the differences between them (red) for both side of the domain. The third panel shows the same, but with convective transport turned off.




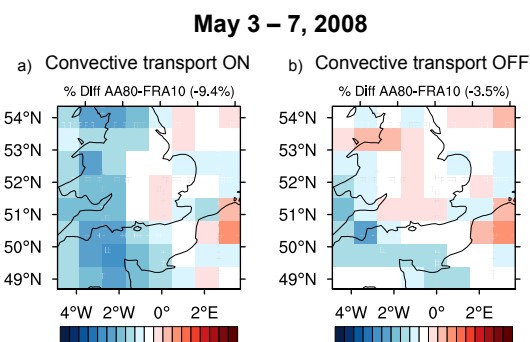

Figure 11: Percent differences in mean spatial distribution of column amount of accumulation mode aerosol water content (in $\mu$g m$^{-2}$) between the FRA10 and AA80 simulations for the period of May 3 - 7, 2008, with convective transport turned on (a) and turned off (b).

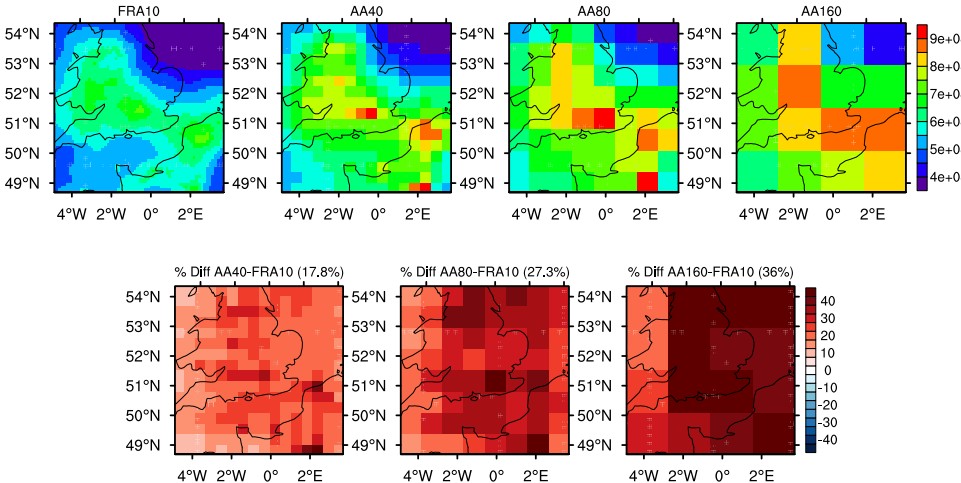

Figure 12: Simulated spatial distribution of CCN at 0.5% supersaturation (in # cm$^{-2}$) for the FRA10, AA40, AA80, and AA160 runs (top row). The bottom row represents the percent difference between each run and FRA10. The number in brackets in the bottom row represents the average percentage difference in the two runs.





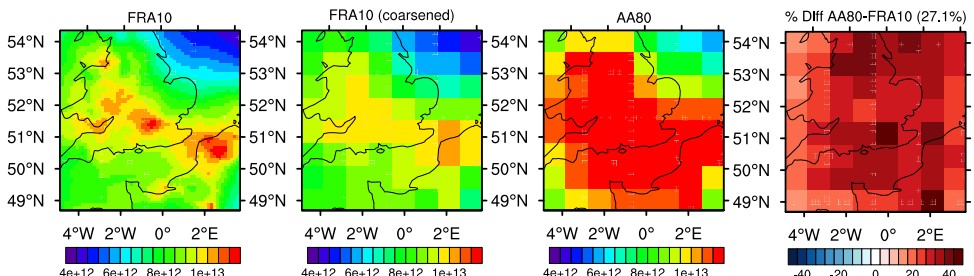

Figure 13: Spatial distribution of column amount of accumulation mode number concentration (# m$^{[-2]}$) for the FRA10 simulation, the FRA10 simulation coarsened to the low resolution grid, the AA80 simulation, and the percent difference between the two. The number in brackets on the third panel represents the average percentage difference in the two runs.

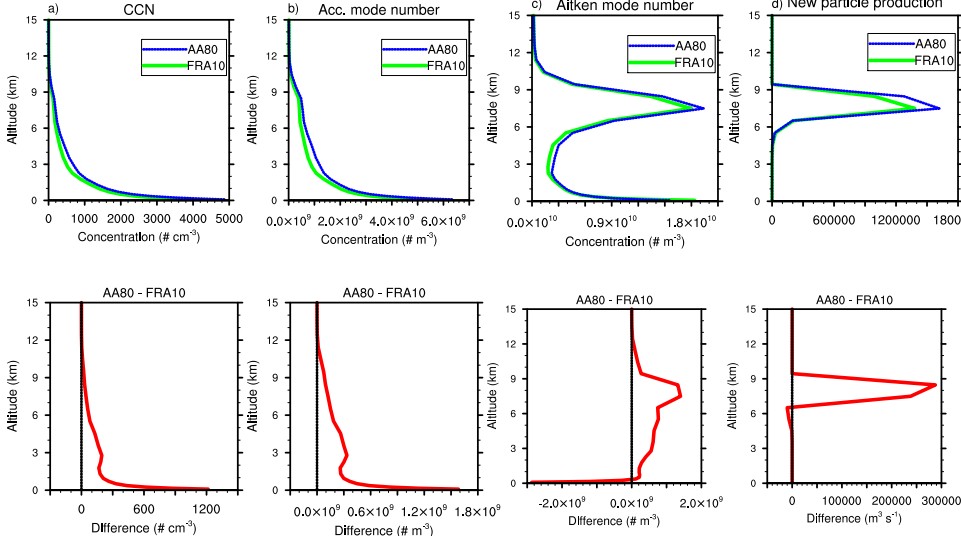

Figure 14: Vertical profiles for the FRA10 and AA80 simulations and the differences between them of CCN at 0.5% supersaturation (a, in # cm$^{-3}$), accumulation mode number concentration (b, in # m$^{-3}$), Aitken mode number concentration (c, in # m$^{-3}$), and nucleation rate (d, in m$^3$ s$^{-1}$). FRA10 simulations are in green, the AA80 in dashed blue and the differences in red.





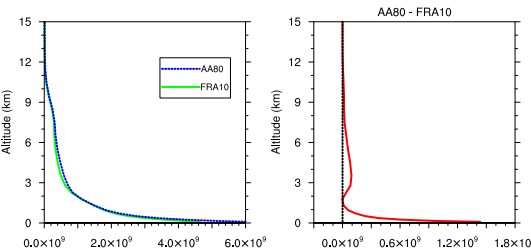

Figure 15: Vertical profile of accumulation mode number concentration (in # m$^{-3}$) from the FRA10 and altered AA80 simulations and the differences between. The altered AA80 simulation averages only the aerosol fields, instead of both gas and aerosol fields.

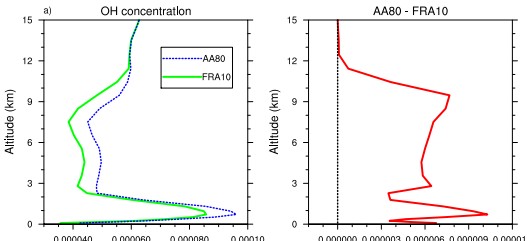

Figure 16: Vertical profiles of OH (in $\mu$g/m$^3$) for the FRA10 and AA80 simulations and the differences between them. FRA10 simulations are in green, the AA80 in dashed blue and the differences in red.

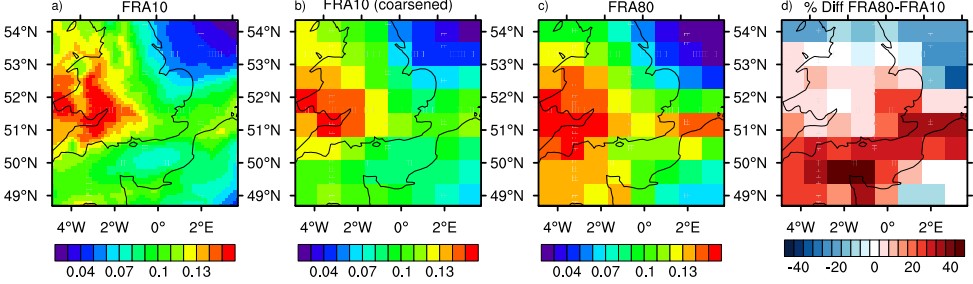

Figure 17: Demonstrates a traditional comparison of the spatial distribution of AOD at two different model resolutions. AOD is simulated at 10 km resolution (a) and 80 km resolution (c). The AOD fields from the higher resolution run are coarsened to the low resolution grid (b) before taking the percent difference between the two runs (d)



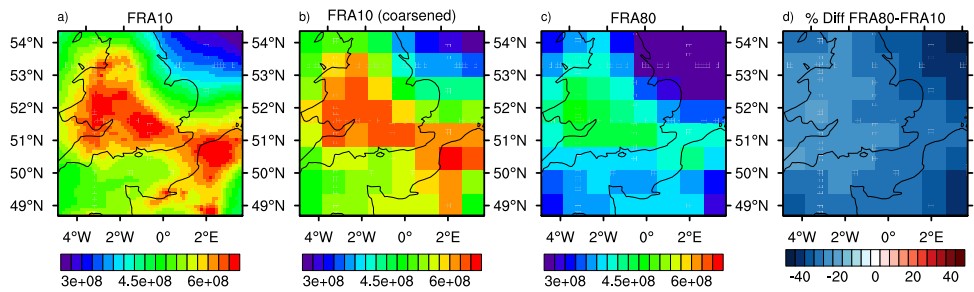

Figure 18: Demonstrates a traditional comparison of the spatial distribution of CCN at 0.5% supersaturation (in # cm$^{-2}$) at two different model resolutions. CCN is simulated at 10 km resolution (a) and 80 km resolution (c). The CCN fields from the higher resolution run are coarsened to the low resolution grid (b) before taking the percent difference between the two runs (d)