# Peer review of "Effect of aerosol sub-grid variability on aerosol optical depth and cloud condensation nuclei: Implications for global aerosol modelling."

_Atmospheric Chemistry and Physics, 2016_

## Referee Comment (RC1) · Anonymous Referee #3 · 22 Jul 2016

In this study, the authors describe a new method that isolates the impact on simulated aerosol and chemistry distributions of changing horizontal resolution of the aerosol and chemistry components. Compared to previous studies looking at resolution impacts, the method guarantees that no other changes, i.e. to the structure of the model or the resolution of other model components, contribute. The causes for the changes can then be fully understood.

Applying their method to WRF-Chem, the authors find that modelled AOD decreases with decreasing resolution because of underestimation of water uptake, itself traced back to the non-linearities of the nitrate equilibrium between gas and aerosol phases, and to convective transport. The authors also find that modelled CCN increases with

decreasing resolution because of non-linearity in nucleation rates, itself traced back, via OH, to ozone production and again convective transport.

The paper is interesting to the aerosol and chemistry modelling communities because it is an in-depth analysis that seeks to understand the roots of differences caused by resolution, and because it convincingly demonstrates that neglecting sub-grid variability has sizeable consequences on weather- and climate-relevant variables like aerosol optical depth and cloud condensation nuclei. The paper is also well written, with well-chosen and good quality figures. For those reasons, I recommend publication.

I would however like to see minor revisions that improve the description of the analysis and discussion in places. I also think that the authors should elaborate on their conclusion that aerosol model development should account for the effects of sub-grid variability.

**1  Main comments**

- The paper's message that sub-grid variability is important and should be accounted for in aerosol model development is well-taken, but also easier to say than do. With their experience of looking into those aspects, the authors must have interesting views on the following questions. Is high resolution required? Line 223 gives an interesting statement in that respect. Do the authors have references or experiences to back up that statement that 10 km is a length scale characteristic of aerosol and CCN distributions? If high resolution is required, does that mean that low resolution simulations should not be trusted? Can low resolution be made to behave like high resolution by imposing subgrid distributions or stochastic parameterisations?

**2 Other comments**

- Lines 134-137 and Figure 2b: it is unfortunate to have chosen to illustrate the effect of inner domain averaging with a variable (ammonia emissions) which does not get averaged in the method. Ammonia surface concentrations would have been a better choice. Having said that, lines 115-120 in the previous section could be interpreted as saying that emissions have also been averaged – but we are now told it is not the case. I guess variables that get passed from module to module are averaged while variables that are only used within one particular module are not. It would be helpful to clearly list in a Table in section 2.1 which variables are averaged in the AA setup.

- Lines 156: How was the length of the spin-up period chosen? Typically spin-up should be long enough for tracer mass budgets to balance for given boundary conditions. 2 days is probably too short, and I am unclear whether both real and averaged aerosol mass budgets should balance, or only the real one.

- Lines 177-181: According to previous sections, averaged variables also include "gases". What are the gases represented in MADE/SORGAM?

- Lines 207-208: So which aerosol types/modes are PM10 emissions emitted into?

- Line 228: Could we have more details on this coarse-graining procedure?

- Lines 236-241: For the sake of completeness, a Table summarising the correlations studied, and the resulting correlation coefficients, would be good.

- Section 3.1.1: The causes for the lack of water uptake in AA80 are well investigated, but there a disconnect between the paragraph discussing ammonia (lines 325-340) and the paragraph discussing vertical profiles (lines 341-354). Should the sentences beginning lines 351 and 359 say that the causal chain begins with

[Figure]

underestimated ammonia in the HRHA regime? That conclusion seems partly confirmed by the discussion in section 3.1.2.

- Table 3: It would be useful to state in the caption that the LRHA regime is not shown.

- Figure 7: What is the unit of x axis?

- Line 459: From this point, the word "mixing" is used to mean averaging or dilution over a grid-box. I am not sure that it is the best use of the word, because of the risk of confusion with vertical mixing. I recommend using averaging instead.

- Line 516: Again, a Table showing the list of variables tested and the resulting correlations would be useful.

- Figures 14 and 15: Figure 15 is only used to make a small point, so its contents could be merged into Figure 14.

**3 Technical comments**

- Line 207: Typo: components

- Line 325: There is a full stop missing somewhere in this sentence.

- Line 449: Extra word: in some

- Caption of Figure 13: Something has gone wrong with square brackets.

- Captions of Figures 14 and 15: Why the plural in "FRA10 simulations"? There is only one FRA10 simulation according to Table 1.

- Line 559: Remove closing bracket.

---

## Referee Comment (RC2) · Anonymous Referee #1 · 21 Aug 2016

The manuscript presents a topic of great interest to the global modeling community. Often it is not clear what are the impact of subgrid-scale variability of aerosol processes.

This study conducts a systematic evaluation separating resolution of aerosol processes, from changes in full model resolution that have other contributing factors e.g. resolution effects due to convection, RH etc.

The paper is well written, the concepts presented are logical and the paper is easy to understand. I recommend publication, after the authors acknowledge some caveats and future directions in their study:

1. In their study the authors hold met and dynamics at 10 km baseline for both averaged

[Figure]

**ACPD**

and high resolution runs. But in global models the met and dynamics are at 80 km. How would their results change if they had used met and dynamics at 80 km baseline?

2. Aerosol nucleation and secondary organic aerosols: Although not the focus of their study aerosol nucleation and new particle formation is significantly affected by low volatility organic vapors (see several recent papers e.g. "Trostl, J., et al. (2016), The role of low-volatility organic compounds in initial particle growth in the atmosphere, Nature, 533(7604), 527"

More future studies similar to what the authors presented are needed not just for in-organic but organic aerosol systems. 3. It should be acknowledged, that the over-estimations in CCN the authors see can get affected by what processes (e.g. effects of organic aerosols, and their non-linear relations with chemistry) are included. The resolution effects, although valuable as presented in their study, are subject to change based on simulations of aerosols and aerosol processes. This caveat is very important to acknowledge in the conclusions sections.

---

## Author Comment (AC1) · 2 Oct 2016

We thank our two reviewers for their helpful comments on our manuscript.

Response to Reviewer 1:

**Main comments**

*The paper's message that sub-grid variability is important and should be accounted for in aerosol model development is well-taken, but also easier to say than do. With their experience of looking into those aspects, the authors must have interesting views on the following questions. Is high resolution required? Line 223 gives an interesting statement in that respect. Do the authors have references or experiences to back up that statement that 10 km is a length scale characteristic of aerosol and CCN distributions? If high resolution is required, does that mean that low resolution simulations should not be trusted? Can low resolution be made to behave like high resolution by imposing subgrid distributions or stochastic parameterisations?*

Our results suggest that high resolution may be required in regimes with complex non-linear thermodynamics. The resolution required may be different in different regimes.

There have been a number of previous studies that have quantified the scales of aerosol variability. For example:

- Anderson et al. (2003) used autocorrelation analysis to show that most of the variation in the aerosol properties existed on scales of 40-160 km.
- Weigum et al. (2012) analysed aircraft measurement of black carbon over the remote Pacific ocean and found that BC variability occurs on scales smaller than 80 – 160 km.
- Targino et al. (2005) performed a similar analysis using autocorrelation functions of aircraft data from clean and polluted regions in the free troposphere. They found the spatial scales of variability to be on the order of 10 km.
- Shinozuka and Redemann (2011) compared the horizontal variability of aerosol optical depth during two contrasting phases of the Arctic Research of the Composition of the Troposphere from Aircraft and Satellites (ARCTAS) campaign. In the first phase, which was dominated by local emission sources, AOD demonstrated considerable variability at scales of 20 km, whereas the second phase, which was dominated by long-range transport, showed very little variability at these scales.

While there is considerable variability in these results, one can see that aerosol variability typically exists on scales in the range of tens of kilometres. We chose to run WRF-Chem at 10 km as this should capture the significant scales of aerosol variability while also being able to use a convective parameterisation (in order to make a more realistic comparison to global models, which typically cannot resolve convection).

There have been previous studies that show that a number of techniques employed to account for aerosol sub-grid variability can lead to improved results (e.g. adaptive grid – Garcia-Menendez et al, 2010; plume-in-grid – Karamchandani et al., 2006; PDFs/stochastic grids – Cassiani et al., 2010). To address this point, I have added the following sentences to the final paragraph of the paper:

"One of the major challenges to future modelling is determining how to account for the sub-grid variability of these aerosol processes. Several methods such as adaptive grid techniques and stochastic parameterisations are being developed to target areas where sub-grid variability is significant. This paper increases our understanding of the underlying mechanisms most affected by sub-grid variability and will help guide future development of these methods in order to more accurately predict the aerosol effect on climate."

**Other comments**

*Lines 134-137 and Figure 2b: it is unfortunate to have chosen to illustrate the effect of inner domain averaging with a variable (ammonia emissions) which does not get averaged in the method. Ammonia surface concentrations would have been a better choice. Having said that, lines 115-120 in the previous section could be interpreted as saying that emissions have also been averaged – but we are now told it is not the case. I guess variables that get passed from module to module are averaged while variables that are only used within one particular module are not. It would be helpful to clearly list in a Table in section 2.1 which variables are averaged in the AA setup.*

To alleviate confusion, I have changed the plot to show ammonia surface concentrations rather than emissions. Additionally, I edited line 115 to make it clearer that emissions are treated as a process rather than fields that could be averaged.

As there are a large number of gases included in the gas-phase chemistry model (RADM2), I have added the sentence below to the description of the model configuration, rather than listing the species in a table.

"Gas-phase atmospheric chemistry is based on the Regional Acid Deposition Model, version 2 (RADM2), which includes 21 inorganic and 42 organic chemical species with 158 reactions, of which 21 are photolytic (Stockwell et al, 1990). These species, along with the aerosols described above, are all averaged in the ``aerosol averaged' simulations."

Edit to line 115: "These processes include emission, photolysis, dry deposition, vertical mixing and wet deposition by convective transport, gas-phase chemistry, and aerosol microphysical processes."

*Lines 156: How was the length of the spin-up period chosen? Typically spin-up should be long enough for tracer mass budgets to balance for given boundary conditions. 2 days is probably too short, and I am unclear whether both real and averaged aerosol mass budgets should balance, or only the real one.*

The concentrations of the aerosol and gaseous species stabilised after two days in both the averaged and unaveraged simulations. We also tested the results using a 5-day spin up period, and there were no differences in the results.

*Lines 177-181: According to previous sections, averaged variables also include "gases". What are the gases represented in MADE/SORGAM?*

See response to first comment.

*Lines 207-208: So which aerosol types/modes are PM10 emissions emitted into?*

PM10 emissions from the TNO inventory remain unspeciated and are emitted into a generic 'PM10' variable. They are coarse mode aerosols.

*Line 228: Could we have more details on this coarse-graining procedure?*

I added the following statement to Line 228:

We calculated the differences by first coarse-graining the results from the high resolution simulation to the grid of the low resolution run to which it is being compared by taking the average of the high resolution output residing within each low resolution cell.

*Lines 236-241: For the sake of completeness, a Table summarising the correlations studied, and the resulting correlation coefficients, would be good.*

This has been added as Table 3.

*Section 3.1.1: The causes for the lack of water uptake in AA80 are well investigated, but there a disconnect between the paragraph discussing ammonia (lines 325-340) and the paragraph discussing vertical profiles (lines 341-354). Should the sentences beginning lines 351 and 359 say that the causal chain begins with underestimated ammonia in the HRHA regime? That conclusion seems partly confirmed by the discussion in section 3.1.2.*

Indeed. The underestimation of nitrate in the BL is likely due to the underestimation in ammonia due to the reasons discussed in lines 325-340. While we cannot conclude this for certain based on the vertical profiles, I have added a statement to line 351 to remind readers of this link.

Edit to line 351: "It is therefore this underestimation in BL nitrate==, which is likely due to the underestimation in ammonia==, that leads to an underestimation in aerosol water content (Figure 6c), and, ultimately, extinction (Figure 6d)."

*Table 3: It would be useful to state in the caption that the LRHA regime is not shown.*

Added: "Results from LRHA are not shown." to caption of Table 3.

*Figure 7: What is the unit of x axis?*

The units of the x-axis are the same as the y-axis. I have added this to the caption to clarify

*Line 459: From this point, the word "mixing" is used to mean averaging or dilution over a grid-box. I am not sure that it is the best use of the word, because of the risk of confusion with vertical mixing. I recommend using averaging instead.*

Good point – I changed 'mixing' to 'averaging'.

*Line 516: Again, a Table showing the list of variables tested and the resulting correlations would be useful.*

This has been added as Table 5.

*Figures 14 and 15: Figure 15 is only used to make a small point, so its contents could be merged into Figure 14. 3*

Figure 15 shows results from a distinct run where only aerosols are averaged, instead of both aerosols and gases. As this comparison is different from the comparisons being made in Figure 14, I chose to keep them separate to alleviate confusion.

**Technical comments**

*Line 207: Typo: components*

Corrected.

*Line 325: There is a full stop missing somewhere in this sentence.*

Corrected.

*Line 449: Extra word: in some*

Corrected.

*Caption of Figure 13: Something has gone wrong with square brackets.*

Corrected.

*Captions of Figures 14 and 15: Why the plural in "FRA10 simulations"? There is only one FRA10 simulation according to Table 1.*

Corrected.

*Line 559: Remove closing bracket.*

Corrected.

---

## Author Comment (AC2) · 2 Oct 2016

We thank our two reviewers for their helpful comments on our manuscript.

Response to Reviewer 2:

1.  *In their study the authors hold met and dynamics at 10 km baseline for both averaged and high resolution runs. But in global models the met and dynamics are at 80 km. How would their results change if they had used met and dynamics at 80 km baseline?*

    This is an interesting point you raise and one that the authors considered. The design of the experiment, however, does not allow for this since we cannot 'create' high resolution data from a low resolution simulation. It is difficult to say how this would affect the results without being able to test it.

2.  *Aerosol nucleation and secondary organic aerosols: Although not the focus of their study aerosol nucleation and new particle formation is significantly affected by low volatility organic vapors (see several recent papers e.g. "Trostl, J., et al. (2016), The role of low-volatility organic compounds in initial particle growth in the atmosphere, Nature, 533(7604), 527" More future studies similar to what the authors presented are needed not just for inorganic but organic aerosol systems.*

    This is true and would be interesting future work. Unfortunately, the nucleation scheme in this version of WRF-Chem includes only the effect of sulphuric acid vapour. This paper is limited in reporting resolution effects of the processes that are included within this specific model – a caveat that you point out in your third comment (I have added a sentence to the conclusion to reflect this). It would be valuable to repeat the experiment with a more sophisticated nucleation scheme to explore what kind of effect this would have on CCN.

3.  *It should be acknowledged, that the overestimations in CCN the authors see can get affected by what processes (e.g. effects of organic aerosols, and their non-linear relations with chemistry) are included. The resolution effects, although valuable as presented in their study, are subject to change based on simulations of aerosols and aerosol processes. This caveat is very important to acknowledge in the conclusions sections.*

    A fair point – I have added the sentence below to Line 713 to reflect this caveat:

    "We should add that while these resolution effects are subject to change based on the aerosol processes that are included in the specific model simulation, these results point to non-linear processes as being most significantly affected."